# SNP- and Haplotype-Based GWAS of Flowering-Related Traits in *Brassica napus*

**DOI:** 10.3390/plants10112475

**Published:** 2021-11-16

**Authors:** MMU Helal, Rafaqat Ali Gill, Minqiang Tang, Li Yang, Ming Hu, Lingli Yang, Meili Xie, Chuanji Zhao, Xiaohui Cheng, Yuanyuan Zhang, Xiong Zhang, Shengyi Liu

**Affiliations:** 1Key Laboratory of Biology and Genetic Improvement of Oil Crops, The Ministry of Agriculture and Rural Affairs, Oil Crops Research Institute of Chinese Academy of Agricultural Sciences, Wuhan 430062, China; mostofa.agr12@gmail.com (M.M.U.H.); drragill@caas.cn (R.A.G.); tangminqiang@hainanu.edu.cn (M.T.); li2.yang@wur.nl (L.Y.); huming199217@126.com (M.H.); yanglingli1120@163.com (L.Y.); xiemeili0101@163.com (M.X.); zhaochuanji@caas.cn (C.Z.); chengxiaohui@caas.cn (X.C.); liusy@oilcrops.cn (S.L.); 2Key Laboratory of Germplasm Innovation of Tropical Special Forest Trees and Ornamental Plants (Ministry of Education), College of Forestry, Hainan University, Haikou 570228, China

**Keywords:** flowering time, *Brassica napus*, SNP, haplotypes, QTL

## Abstract

Traits related to flowering time are the most promising agronomic traits that directly impact the seed yield and oil quality of rapeseed (*Brassica napus* L.). Developing early flowering and maturity rapeseed varieties is an important breeding objective in *B. napus*. Many studies have reported on days to flowering, but few have reported on budding, bolting, and the interval between bolting and DTF. Therefore, elucidating the genetic architecture of QTLs and genes regulating flowering time, we presented an integrated investigation on SNP and haplotype-based genome-wide association study of 373 diverse *B. napus* germplasm, which were genotyped by the 60K SNP array and were phenotyped in the four environments. The results showed that a total of 15 and 37 QTLs were detected from SNP and haplotype-based GWAS, respectively. Among them, seven QTL clusters were identified by haplotype-based GWAS. Moreover, three and eight environmentally stable QTLs were detected by SNP-GWAS and haplotype-based GWAS, respectively. By integrating the above two approaches and by co-localizing the four traits, ten (10) genomic regions were under selection on chromosomes A03, A07, A08, A10, C06, C07, and C08. Interestingly, the genomic regions FT.A07.1, FT.A08, FT.C06, and FT.C07 were identified as novel. In these ten regions, a total of 197 genes controlling FT were detected, of which 14 highly expressed DEGs were orthologous to 13 *Arabidopsis thaliana* genes after integration with transcriptome results. In a nutshell, the above results uncovered the genetic architecture of important agronomic traits related to flowering time and provided a basis for multiple molecular marker-trait associations in *B. napus*.

## 1. Introduction

Oilseed rape (*Brassica napus* L.) is ranked third in world vegetative oil production (USDA ERS, 2019; https://www.ers.usda.gov/data-products/oil-crops-yearbook/ accessed on 11 May 2021). It is mainly consumed as an edible oil for humans and is also used as an industrial material for lubricants and biodiesel production. *B. napus* (genome AACC, 2n = 38) is a young allopolyploid species that arose from the spontaneous interspecific hybridization between *B. rapa* (genome AA, 2n = 20; also called Asian cabbage, turnip) and *B. oleracea* (genome CC, 2n = 18; European cabbage) at least 10,000 years ago following chromosome doubling [1,2]. The rice–rice–rapeseed (RRR) rotation is a mainstay of China’s food security and economy. *B. napus* was introduced to China from Europe in the 1930s–1940s, but rapeseed’s increasing growth period poses challenges to this rotation system [3,4]. Therefore, breeding early maturing rapeseed varieties are the best way to solve this problem.

Flowering is a key component of a plant’s life cycle that marks its transition from the vegetative to reproductive phase. Opportune flowering time is crucial for survival in specific environments and regulates life-cycle duration, yield, seed quality, resistance to diseases, and crop rotation systems. Flowering time (FT) is a dynamic quantitative trait that is regulated by a complex network and responds to endogenous and exogenous stimuli, such as photoperiod, temperature, and hormones [5]. So far, six major regulatory pathways, the vernalization, photoperiod, gibberellic acid (GA), autonomous, thermal clock, and aging pathways, are involved in flowering time and have been identified in the model plant *Arabidopsis thaliana*. A systematic overview of FT’s molecular mechanisms has been extensively investigated in *A. thaliana*, and more than 300 genes in the above six pathways are involved in the flowering process [6]. Vernalization and photoperiod pathways play a major role in shaping flowering time diversity in different crop plants. *FLOWERING LOCUS C (FLC)* and *FRIGIDA (FRI)* are essential genes in the vernalization response in *Arabidopsis*, whereas *CONSTANS (CO)* is engaged in photoperiod response [7]. In addition, *FLOWERING LOCUS T (FT)* contains a mobile signal termed “florigen,” which acts as a major floral integrator in flowering control [8].

In recent years, owing to advances in bioinformatics and sequencing technology, the genetic dissection of complex crop traits has been achievable. Genome-wide association studies (GWAS) were first applied in the study of human diseases as a cost-effective tool [9]. Then, they were successfully applied to dissect genetic architecture and to identify the potential candidate genes underlying the quantitative trait locus (QTLs) of important agronomic traits from diverse germplasm with high-density single nucleotide polymorphisms (SNPs) [10]. In *B. napus*, GWAS has been widely used to mine the genetic variants that are associated with crop agronomic traits, including FT-related traits using the Illumina Infinium *Brassica* 60K SNP array [11,12]. For example, Xu et al. (2016) identified 35 FT loci with a diversity panel comprising 523 inbred rapeseed lines [11]. Raman et al. (2016) also found 69 SNP markers that were linked with flowering time and numerous putative flowering time genes within 20 Kb areas, including *FT*, *FRUITFUL*, *FLC*, *CO*, *FRI*, and *PHYB* [13]. A total of 131 and 146 SNP loci were associated with FT traits using GWAS based on SNP by Zhou et al. (2018) [14] and Wang et al. (2020) [15], respectively. Li et al. (2018) reported a total of 55 QTLs related to flowering time through a doubled haploid (DH) population of 348 lines and also showed a genetic linkage map along with SNP markers [16]. Jian et al. (2019) reported that 27 FT QTLs were distributed on eight chromosomes [12]. Recently, haplotype-based GWAS (hap-GWAS) have shown promise in barley [17], wheat [18], and maize [19]. Haplotype construction compresses multiple SNPs into a haplotype locus and optimizes the design of genomic selection (GS) and GWAS [20]. Sehgal et al. (2020) reported seven, four, ten, and fifteen stable haplotype associations in wheat in irrigated, moderate drought, extreme drought, and heat stress testing conditions, respectively [19]. Therefore, understanding the genetic mechanisms of flowering time traits under multiple environments is one of the important aspects in plant breeding. During the transition from the vegetative to the reproductive stage, the flowering of a plant followed a sequence of some developmental incidents such as bud formation, transition from stem to inflorescence meristem, floral development, and flower blooming, which were depicted as budding, bolting, and days to flowering (DTF) in *B. napus*. Further, the days between bolting and DTF can be quantified and described as IBD (interval between bolting and DTF). However, these research studies mainly focused on DTF, with only a few focusing on the budding and bolting of *B. napus* and the QTLs associated with the interval trait (IBD) identified only in *Arabidopsis* [21]. Analyses of these traits using SNP-based GWAS combined with hap-GWAS as well as testing under multiple environmental conditions (MEC) have not been reported previously. Crop plants with diverse backgrounds and that are grown under MEC can potentially detect environmentally stable loci and thencandidate genes controlling commercially important traits [22]. Interestingly, *B. napus* are able to adapt to different geographical conditions well by adjusting their FT, i.e., winter, semi-winter, and spring types [23,24]. Additionally, combining transcriptomics allows GWAS to identify natural variation and the genetic basis of phenotypes effectively.

RNA sequencing has been commonly used to investigate variations in global gene expression. A total of 105 FT DEGs were identified by Jian et al. (2019) [12]. In addition, transcriptome research combining DEGs with GWAS or linkage mapping has been thought to be an efficient way to find candidate genes for complex traits [25]. By integrating GWAS and linkage mapping with DEGs [12,15], Jian et al. (2019) and Wang et al. (2020), found 8 and 12 FT genes, respectively, in total.

As such, knowing about the molecular mechanism and identifying the functional markers of budding, bolting, DTF, and IBD traits in *B. napus* using the 60K SNP array of 373 diverse accessions that are directly related to the flowering process is one of the most important breeding targets in *Brassica* breeding. In addition, we also used the shoot apical meristem (SAM) tissue of two early and late flowering accessions to perform RNA-Seq analysis. The main objectives of our study are the (i) identification of environmentally stable, environmentally specific, pleiotropic novel genetic loci/QTLs and candidate genes regulating FT-related traits through SNP and hap-GWAS; (ii) the dissection of early and late FT DEGs of SAM using transcriptomics study; and (iii) the identification of FT candidate genes through combining association mapping with DEGs for further study. Our results will provide a comprehensive overview of the genetic and molecular mechanisms underlying FT traits and new molecular markers for breeders to develop superior rapeseed varieties.

## 2. Results

### 2.1. Phenotypic Variation and Correlations of Measured Traits Related to FT of B. napus

A total of 373 worldwide diverse *B. napus* accessions (Appendix A) were used for GWAS. The phenotypic variations for budding, bolting, DTF, and IBD of 373 rapeseed accessions were measured for three replications in two different locations and are presented in Table 1. For example, the DTF of these accession means ranged from 162.49 days to 167.89 days across four environments and had a minimum coefficient of variation of 2.64% to 7.51% (Table 1). Additionally, with the exception of budding in 2013WH, each of the four traits observed in four environments were shown to be approximately continuous and normally distributed, indicating that the group was acceptable for association analysis (Figure 1). For all four variables, there were significant differences between environments and genotypes. Analysis of variance (ANOVA) was used to test the influence of genotype (G), environment (E), and their interactions (G × E) for the four flowering traits (Appendix A). Additionally, these four traits showed significant G × E interaction. In all contexts, budding and bolting have the highest broad-sense heritability (*h*^2^) of >90%, with the exception of 2013YZ, but IBD has low heritability. The broad-sense heritability (*h*^2^) value of budding, bolting, DTF, and IBD was mostly greater than 80% across 2013WH, 2014WH, and 2015WH, but not 2013YZ, indicating that these traits may be affected by the environment to a certain extent. Overall, these results showed that all four traits were inherited in a stable manner.

Our findings demonstrate that based on the BLUE value results (Figure 2), the paired phenotypic correlations between the traits related to FT were highly significant (*p* < 0.001). Budding showed significant positive correlations with bolting, DTF showed significant correlation coefficients of 0.71 and 0.63, and in contrast, budding showed significant negative correlations (*r* = −0.62) with IBD. Bolting was significantly correlated with DTF (*r* = 0.91) and significant negative correlations with IBD (*r* = −0.85). DTF showed significant negative correlations with IBD, with correlation coefficients of −0.55. These results suggested that these four traits were not independently regulated.

### 2.2. SNP-GWAS Uncovers the Candidate QTLs Targeting Traits Related to FT in B. napus

In this study, 21,856 SNP markers were employed after filtering from 60K array data of 373 *B. napus* accessions to perform GWAS for four FT traits using the MLM model. In the current study, to identify the environmentally stable and specific QTLs, we used the phenotypic data of four FT traits grown under four different environments, such as Wuhan 2013, Wuhan 2014, Wuhan 2015 (WH13, WH14, and WH15), and Yangzhou 2013 (YZ13), representing the major oilseed rape (OSR) growing regions in China.

The SNP-GWAS approach revealed a total of 61 significant SNP loci (Figure 3 and Appendix A) associated with four traits related to FT across all four environments and constituted the identification of 15 QTLs distributed on ten chromosomes of A01, A02, A03, A07, A10, C01, C06, C07, C08, and C09 of *B. napus* (Appendix A). Among these 15 QTLs, 3 were detected in at least 2 environments, including qFT.A07.1 (*p* = 2.82 × 10^−5^) located on linkage group A07, which was linked to 2 phenotypic traits (budding and bolting). The second QTL qFT.A10 was located on chrA10 (*p* = 7.02 × 10^−8^) and targeted bolting and DTF in all environments. The third one, qFT.C06, was located on chrC06 (*p* = 5.31 × 10^−6^) target budding in WH14 and YZ13. Overall, these three QTLs showed environmental stability, which suggested that these most significant QTLs should be given full consideration for flowering time marker-trait associations (MTA) for *B. napus*.

Our results also showed that qFT.A01.1 (chrA01: 0.358 Mb) targeting budding possessed an essential gene controlling FT, such as *FLOWERING LOCUS D* (*FD*, *BnaA01g01640D*), which has not been reported to date; thus, it is speculated to be a novel QTL. Another two QTLs, viz. qFT.C08 (3.024 Mb) and qFT.C09 (45.874 Mb), were also found in this study, which was a potentially regulated DTF. Here, qFT.C08 was not discovered yet, whereas qFT.C09 was already illustrated by a previously reported study [14,26,27].

### 2.3. Hap-GWAS Explored QTLs Targeting Traits Related to FT in B. napus

According to Ma et al. (2018) the conventional SNP-based GWAS approach has a high rate of false-negative outcomes, and due to the existence of a significant number of loci with minor effects, the identification of candidate genes linked to complex phenotypic traits is still challenging [28]. Compared to hap-GWAS, SNP-GWAS had a low resolution to identify more candidate regions. Therefore, we also used hap-GWAS in our study to solve this limitation, with a total of 4451 haplotypes (4.91 SNPs/haplotype). In total, 80 putative quantitative trait haplotypes (QTH) were promoted (Figure 4, Appendix A), with a suggestive significance threshold of *p* = 3.65 × 10^−4^ [*p* = −log_10_(1/4451)].

A total of 37 hap.QTLs (Appendix A) were associated with FT traits and were distributed to all chromosomes, except for C09 of OSR across, all environments. Among them, eight hap.QTLs, viz. hap.qFT.A03.2, hap.qFT.A05.1, hap.qFT.A07.2, hap.qFT.A08, hap.qFT.A10.3, hap.qFT.C01.1, hap.qFT.C04.2, and hap.qFT.C08, were co-detected in at least two environments. The hap-QTL hap.qFT.A07.2 and hap.qFT.C01.1 were detected in two different environments and controlled budding and IBD, which suggests that these most momentous regions show environmental stability and pleiotropy and should be focused on to improve the phenotypes for future breeding purposes (Appendix A).

On the other hand, five environmentally repetitive hap.QTLs, viz. hap.qFT.A05.1, hap.qFT.A08, hap.qFT.A10.3, hap.qFT.C04.2, and hap.qFT.C08, were identified in at least two consecutive years in the same location, showing environmental specificity (Appendix A).

Interestingly, hap.qFT.A03.2 regulated all of the flowering traits and showed pleiotropy, environmental stability, and specificity. These results indicated that the climate had a substantial effect on the flowering traits of *B. napus* and that it should be considered in its entirety in order to improve phenotypes for future breeding purposes.

#### Hap-GWAS Revealed QTL Hotspots in *B. napus*

A QTL cluster/hotspot is a densely populated QTL/genomic region of the chromosome containing many QTLs linked to multiple traits in multiple environments. It is important to look at co-localized QTLs from a breeding perspective when considering phenotypic and genetic similarity.

Hap-GWAS detected seven QTL clusters in multiple environments across the linkage groups A03, A05, A06, A09, A10, C02, and C06. For example, a major QTL hotspot (hap.qFT.A03.1, hap.qFT.A03.2, and hap.qFT.A03.3) was identified in the traits spanning ~23.59 Mb on linkage group A03 in different environments. Another, major QTL hotspot (hap.qFT.C06.1, hap.qFT.C06.2, hap.qFT.C06.3, and hap.qFT.C06.4) spanning approximately ~23.46 Mb on chromosome C06 was identified for all traits (Appendix A). Another, five major QTL hotspots were obtained on linkage groups A05, A06, A09, A10, and C02 and were associated with all of the phenotypes spanning from approximately ~17.39 Mb, ~21.39 Mb, ~16.73 Mb, ~15.89 Mb, and ~34.73 Mb, respectively (Appendix A).

### 2.4. Integration of SNP and Hap-GWAS with the Co-Localization of Traits Related to FT

Lu et al. 2011 suggested that using a combination of SNP and hap-GWAS to classify associated signals is preferable compared to a single approach, which always gives precise information about the genomic regions that are associated with phenotypes [29]. By integrating SNP and hap-GWAS, seven genomic regions were found to be distributed on chromosomes A03, A07, A10, C06, and C07 and co-localized by at least two traits, with the exception of FT.C06.1, as shown in Appendix A. In order to comprehensively detect the loci controlling FT, the genomic regions targeting budding, bolting, DTF, and IBD were consistent in both GWAS approaches and were also retained as co-localized QTLs. A total of three co-localized genomic regions (FT.A03.2, FT.A08, and FT.C08) were also identified (Appendix A). Finally, a total of 197 FT genes (Appendix A) were identified in these ten genomic regions within one Mb flanking region. Moreover, the functions of these genes were annotated based on the FLOR-ID [6] and TAIR databases. To confirm the target genomic regions, already reported FT genomic loci based on the “Darmor-bzh” reference genome [2] were evaluated. There were specific differences in the loci of FT among different research backgrounds but within certain concordant intervals. A total of 533 loci linked to FT were extracted, and they were found on almost all of the chromosomes. The linkage groups A02, A03, A07, A09, A10, C01, C02, C03, C04, and C09 have many loci (394/533) (Appendix A).

#### 2.4.1. GWAS Identified Important Known Genomic Regions Controlling FT in *B. napus*

SNP-based GWAS detected 15 QTLs, in which a total of four were detected as known QTLs. On the other hand, from the hap-GWAS data, we also identified a total of 12 known hap.QTLs out of 37 hap.QTLs. After combining the SNP and hap-GWAS by considering the co-localized QTLs, six QTLs were revealed as known genomic regions that were distributed on the linkage groups A03, A07, A10, and C08, which are presented in Appendix A. Among the known genomic regions, the FT.A03.1 (chrA03: 5.461–7.355 Mb), FT.A03.3 (chrA03: 26.381–27.390 Mb), FT.A07.2 (chrA07: 22.475–23.971), and FT.A10 (chrA10: 13.003–15.950 Mb) regions were detected in both association approaches, and each regulated at least two traits.

The genomic regions FT.A03.1 and FT.A10 were detected at the proximal side of the A03 and A10 chromosomes, respectively. FT.A03.1 was discovered by a previously reported study [16,23,27,30,31,32,33] that harbored 21 FT genes, including the central flowering time gene *FRIGIDIA* (*FRI)-BnaA03g13320D* and *FLOWERING LOCUS C (FLC)-BnaA03g13630D*. Remarkably, the most stable FT.A10 region [11,12,14,16,30,31,32,33,34] out of 32 FT genes, identified a few of the most important central FT genes, including *BnaA10g18010D* (*FRIGIDA like 1-FRL1*), *CONSTANS* (*CO*)-*BnaA10g18430D*, and *FLOWERING LOCUS C* (*FLC*)-*BnaA10g22080D*, and regulated bolting, DTF, and IBD. Of these, the essential winter annual flowering gene *BnaA10g18010D* (*FRIGIDA like 1-FRL1*) and a MADS-box protein *FLOWERING LOCUS C (FLC)* ortholog of *BnaA10g22080D* functioned as a repressor of floral transition [35] as well as a photomorphogenesis gene, *CONSTANS (CO)-BnaA10g18430D*, that encodes a protein that resembles zinc finger transcription factors and is involved in flowering control during long days [36].

Another two integrated regions, FT.A03.3 (chrA03: 26.381–27.390 Mb) and FT.A07.2 (chrA07: 22.475–23.971), were identified on the distal side on the linkage groups A03 and A07, respectively. FT.A03.3 possessed 13 FT genes, which regulated bolting, DTF, and IBD and possessed the essential flowering gene *AGAMOUS-like 16 (BnaA03g51000D*-*AGL16*), a MADS-box transcription factor, and *FPF1-like protein 1* (*BnaA03g51440D*-*FPL1*), which encodes a small protein and is similar to floral promoting factor-like 1. A total of 20 FT genes with the essential flowering gene *TWIIN SISTER FLOWER (TSF)*-*BnaA07g33120D* were found to be detected in FT.A07.2 and were discovered by An et al. (2019), Wang et al. (2020), and Wu et al. (2019) [15,32,37]. These core findings suggested further breeding improvement of the floral transition in *B. napus.*

#### 2.4.2. Identification of Novel Genomic Regions Targeting Traits Related to FT in *B. napus*

A total of eleven and twenty-five novel QTLs were identified through SNP and hap-GWAS, respectively. Four unexplored regions were merged by both SNP and hap-GWAS and trait co-localization on chromosomes A07, A08, C06, and C07. The genomic regions FT.A07.1 (chrA07: 12.369–14.660 Mb), FT.C06 (chrC06: 24.474–26.439 Mb), and FT.C07 (chrC07: 42.132–42.329 Mb) were the result of the integration of both approaches. The co-localized region FT.A08 was detected by hap-GWAS and control bolting, DTF, and IBD. A total of 18 and 17 FT genes were identified in the FT.A07.1 and FT.C06 region, including *cycling DOF factor 2 (CDF2)-BnaA07g14740D* and *PEPPER (PEP)-BnaC06g24760D*, respectively, which was not reported by any of the previously published studies, and speculated as to be a novel region. The other two novel regions were also regulated at least two phenotypes. Interestingly, these two novel regions viz. FT.A08 (chrA08: 1.419–3.281 Mb) and FT.C07 (chrC07: 42.132–42.329 Mb) controlled both bolting and IBD. However, it was hypothesized that these integrated and co-localized novels and stable genomic regions were retained for further study to comprehensively classify the flowering time loci in *B. napus.*

In a nutshell, these results suggest that using both SNP and hap-GWAS to detect natural variation influencing target traits could increase the detection accuracy and efficiency. These QTLs and genes are thought to be reliable candidate genes for mediating flowering traits in *B. napus*, and further confirmation of their role would aid in elucidating the genetic and molecular mechanisms underlying yield-related traits.

### 2.5. Transcriptomics Analysis

To identify the DEGs related to flowering time, we sequenced two early samples, AH110 (SE1) and AH275 (SE2), and two late samples, AH218 (SL1) and AH245 (SL2), of SAM tissue with two biological replications. There was a total of 12.31 Gb (AH110D), 15.27 Gb (AH110E), 12.11 Gb (AH275B), 10.06 Gb (AH275C), 12.24 Gb (AH218B), 12.63 Gb (AH218C), 12.79 Gb (AH245B), and 11.41 Gb (AH245C) clean bases acquired in each sample (Appendix A). Mapping with the *B. napus* reference genome, total mapping reads, and clean reads are shown in Appendix A. The correlation coefficient (R^2^) among the samples based on FPKM (fragments per kilobase transcript per million reads) values showed the reliability of the biological replications ensuring the integrity of transcriptome data and were used for subsequent analysis (Appendix A).

A false discovery rate (FDR) < 0.05 and log_2_FC > 1 was used as the threshold to judge the DEGs between the early and late flowering samples. A total of 11608 DEGs were found between SE1 vs. SL1 (6111 up-regulated and 5497 down-regulated), 13066 DEGs were identified between SE1 vs. SL2 (6535 up-regulated and 6531 down-regulated), 13,042 DEGs were detected between SE2 vs. SL1 (5537 up-regulated and 7505 down-regulated), and 14494 DEGs were obtained between SE2 vs. SL2 (5986 up and 8508 down-regulation) (Appendix A).

#### Identification of DEGs Related to Flowering Time

According to the BLASTN analysis, we identified a total of 1192 highly expressed homologs of FT DEGs in the *B. napus* (Appendix A). A total of 595, 635, 659, and 720 FT DEGs were detected compared to SE1 vs. SL1, SE1 vs. SL2, SE2 vs. SL1, and SE2 vs. SL2, respectively. In addition, 305 and 362 FT DEGs were found to be common between SE1 vs. SL1/SL2 and SE2 vs. SL1/SL2, respectively (Appendix A). These FT genes are mainly included in the photoperiod, circadian clock, vernalization, gibberellin, aging, autonomous, and ambient temperature pathways.

Among all the 1192 FT genes, some of the important Arabidopsis homologs, such as *CONSTANS (CO)*, *CONSTANS-like 9 (COL9)*, *CRYPTOCHROME-INTERACTING BASIC-HELIX-LOOP-HELIX 1 (CIB1)*, *cycling DOF factor 1-3 (CDF1-3)*, *EARLY FLOWERING 3 (ELF3)*, *EARLY FLOWERING 4 (ELF4)*, *LATE ELONGATED HYPOCOTYL (LHY)*, *PHYTOCHROME INTERACTING FACTOR 4 (PIF4)*, etc., were found to be included in the photoperiod pathway. The circadian rhythm is an essential part of the photoperiod pathway and regulates plant flowering, which acts as an internal timekeeper and regulates daily and seasonal changes [38]. In the circadian clock pathway, some major genes such as *Circadian clock associated 1 (CCA1)*, *TIMING OF CAB EXPRESSION 1 (TOC1)*, *pseudo response regulator 5 (PRR5)*, *pseudo response regulator 7 (PRR7)*, *pseudo response regulator 9 (PRR9)*, etc., were identified. Most of the important flowering time integrators, such as *FLOWERING LOCUS C (FLC)* and *FRIGIDA (FRI)*, were found in the vernalization pathway. Some of the other genes such as *AGAMOUS like 19 (AGL19) EMBRYONIC FLOWER 2 (EMF2)*, *FRIGIDA like 1 (FRL1)*, *VERNALIZATION INSENSITIVE 3 (VIN3)*, *REDUCED VERNALIZATION 1 (VRN1)*, *VERNALIZATION 2 (VRN2)*, etc., were also detected in the vernalization pathway. For the aging pathway, some important genes such as *SQUAMOSA PROMOTER BINDING PROTEIN-LIKE 9 (SPL9)*, *SQUAMOSA PROMOTER BINDING PROTEIN-LIKE 10 (SPL10)*, *SQUAMOSA PROMOTER BINDING PROTEIN-LIKE 15 (SPL15)*, *TARGET OF EARLY ACTIVATION TAGGED 1 (TOE1)*, *TARGET OF EAT 2 (TOE2)*, *TREHALOSE-6-PHOSPHATE SYNTHASE 1 (TPS1)*, *SEPALLATA 1 (SEP1)*, *SEPALLATA 2 (SEP2)*, *SEPALLATA 3 (SEP3)*, etc., were also found. We also found GA receptor *GA INSENSITIVE DWARF1A (GID1A)*, *GA INSENSITIVE DWARF1B (GID1B)*, *GA2 oxidase 1 (GA2ox1)*, *GA2 oxidase 3 (GA2ox3)*, *gibberellin 20-oxidase 3 (GA20OX3)*, *DELLA protein RGA-like 1 (RGL1)*, *RGA-like 2 (RGL2)*, *RGA-like 3 (RGL3)*, etc., which are *Arabidopsis* homolog genes that are related to the gibberellin signaling pathway. These FT DEGs will be important for the future flowering time gene expression study in *B. napus*.

### 2.6. Prioritization of Potential Candidate Genes for FT by Integrating SNP and Hap-GWAS and Co-Localized Traits with DEGs

According to Li et al. (2020), a combination of association mapping with DEGs is a powerful and popular approach for identifying potential candidate genes of target traits [39]. In our study, after integrating ten integrated co-localized regions with highly expressed DEGs results, a total of 14 highly expressed FT-related genes (Table 2 and Figure 5) were selected in four genomic regions. Gene annotation data suggested that most of these above genes are annotated.

For example, some essential annotated genes were *FRIGIDA* (*FRI*), an ortholog of *BnaA03g13320D*; *C-repeat/DRE binding factor 1* (*CBF1*), an *ortholog of BnaA03g13620D*; *FLOWERING LOCUS C* (*FLC*), an ortholog of *BnaA03g13630D* in FT.A03 region; *RELATED TO AP2.7* (*RAP2.7*), an ortholog of *BnaA07g13990D*; *cycling DOF factor 2* (*CDF2*), an ortholog of *BnaA07g14740D*, and S*quamosa Promoter binding protein-like 15* (*SPL15*), an ortholog of *BnaA07g17550D* in linkage group *A07*; *CONSTANS-like 1* (*COL1*) ortholog of *BnaA10g18420D*; *SEPALLATA1* (*SEP1*), an ortholog of *BnaA10g18480D*; *FLOWERING LOCUS C* (*FLC*), an ortholog of *BnaA10g22080D*, and *Ubiquitin-specific protease 12* (*UBP12*), an ortholog of *BnaA10g24300D* in locus FT.A10; *ANTHOCYANIN11* (*ATAN11*), an ortholog of *BnaC08g40840D*; *DWARF AND DELAYED FLOWERING 1* (*DDF1*), an ortholog of *BNAC08G41070D*; *Phytochrome A *(*PHYA*), an ortholog of *BnaC08g42660D*; and *Nuclear factor Y*, *subunit C10* (*NF-YC10*), an ortholog of *BnaC08g43430D* in the genomic region FT.C08.

## 3. Discussion

### 3.1. Importance of Diverse Germplasm for Agronomic Traits Related to FT

The agronomic traits related to FT are the most important traits that directly impact the yield and seed quality of crops. Therefore, the identification of novel loci controlling phenotypes related to FT could aid in elucidating the genetic basis for the development of cultivars adapted to different geographical regions. In the present study, we identified several significant SNP loci and favorable haplotypes through SNP and hap-GWAS approaches, respectively, that regulate traits related to FT under four environments in a 373 diverse panel of *B. napus* natural population.

### 3.2. The Implication of GWAS in Genomic-Assisted Breeding of B. napus

Understanding the genetic variation of different FT-related traits helps to develop early flowering cultivars. By obtaining these genetic variations across the whole genome, GWAS is one of the most powerful techniques for distinguishing the genetic basis of complex traits [40] and provides valuable information for crop breeding improvement. The performance of associative transcriptomics [41] and the 60K SNP chip [42] in detecting the genetic loci-influencing traits in rapeseed suggested that GWAS is an efficient approach for excavating the genomic regions in rapeseed. To date, numerous genetic loci have been identified in *B. napus* through GWAS linked with plant height [43], primary branch number [43], yield-related traits [44], flowering time [11], sclerotinia stem rot resistance [45], etc. In *B. napus*, genome-wide association (GWA) mapping is a highly successful gene discovery method, with a significant portion of phenotypic variation clarified by a small number of quantitative trait loci (QTLs). In our study, we identified a total of 15 QTLs in SNP GWAS and 37 hap.QTLs in hap-GWAS, few of which overlapped with previously reported results.

For instance, in SNP-GWAS, one of the most stable QTL “qFT.A02.2” overlapped with the previously reported results [11,14,15,16,26,30,32] and harbored the central flowering integrator *FLOWERING LOCUS T (FT)-BnaA02g12130D*. Remarkably, several environmentally stable and specific hap.QTLs were also detected in this study, which was previously associated with *B. napus* flowering time. For example, environmentally specific hap.qFT.A03.2 controlled the four FT traits, which have already been explored by An et al. (2019) [37]; N. Wang et al. (2016) [30]; Wang et al. (2020) [15]; and Zhou et al. (2018) [14] and were found to be associated with four phenotypes and to be harboring 25 FT genes, including *AGAMOUS (AG)-BnaA03g43820D*, which may be helpful for the transition of flowering from the vegetative to reproductive phase. These findings strongly support the GWAS results and increase the reliability of the trait-associated SNP loci identified in our study. Hence, based on our results, breeders can directly obtain valuable data and resources for further research in rapeseed.

### 3.3. SNP and Hap-GWAS Co-Localized QTLs Suggested a High Degree of Pleiotropy in Controlling Traits Related to FT in Rapeseed

The term pleiotropy refers to regulating several traits by a single gene or variant [46]. Pleiotropy appears to be elusive in reverse genetics and QTL mapping approaches in different species [47], but it is a common phenomenon in the GWAS mapping of a diverse germplasm of crops [48,49]. According to Wang et al. (2020) [15], 5 and 15 QTLs were identified in the GWAS, and QTL mapping was co-localized with at least two growth period traits in *B. napus.* A total of 4 pleiotropic SNP QTLs and 12 hap.QTLs were discovered to be associated with traits related to FT in our research (Appendix A). In total, six co-localized genomic regions were also be classified as pleiotropic between the integration of SNP and hap-GWAS. The genomic regions FT.A03.1, FT.A03.3, FT.A07.1, FT.A07.2, FT.A10, and FT.C07 were associated with at least two traits. These pleiotropic QTLs discovered the ability to control multiple flowering traits, which helps us to understand the molecular mechanisms of flowering time in *B. napus*.

According to Auge et al. (2019) [50], some FT genes also have been shown to have pleiotropic functions during development. For example, *FLOWERING LOCUS C (FLC)* is a MADS-box transcription factor that functions as a flowering repressor in *Arabidopsis* [35]. It is also involved in germination, an essential stage in plant growth [51] and vegetative development [52]. In this research, the master copy of *FLC (BnaFLC.A10)* also played a pleiotropic role and was correlated with vegetative (bolting), reproductive (DTF), and interval (IBD) traits. This result further indicated that single gene pleiotropy induces QTL co-mapping. In marker-assisted breeding, QTL co-mapping will help breeders to identify favorable alleles for multiple traits simultaneously.

### 3.4. Comparison of SNP and Hap-GWAS Approaches to Identify QTLs Targeting Traits Related to FT in B. napus

SNPs have a limited number of alleles per locus, reducing the polymorphism information content (PIC) for each locus, while haplotype data has several haplotypes (2–9) per locus and a higher degree of allele diversity [53]. According to Lorenz et al. (2010), hap-GWAS found more loci that were associated with barley heading dates than single SNP approaches [54]. As a result, we discovered that the conventional single-SNP model showed fewer associations than a haplotype-based approach. SNP-GWAS, for example, found 61 loci associated with FT traits, while hap-GWAS found 80 significant associations in this study.

According to Contreras-Soto et al. (2017) [55] and N’Diaye et al. (2017) [56], hap-GWAS identified QTLs that were not captured by a single SNP marker system. SNP and hap-GWAS identified 15 and 37 QTLs, respectively, in our research. Notably, seven major QTL clusters were associated with phenotypes in hap-GWAS, while none were in SNP-GWAS. Compared to the single SNP system, Contreras-Soto et al. (2017) reported that haplotypes-approached association may provide new intuitions into the genetic basis of traits [55]. In hap-GWAS, for example, 25 novel QTL were discovered to be correlated with four flowering traits, while only eleven were found in SNP-based GWAS in our research. Furthermore, Wang et al. (2016) and Wen et al. (2018) found that multi-locus methods detected more new genes and fewer existing genes for traits related to flowering in *A. thaliana* than single-locus methods [49,57]. As a result, single SNP and hap-GWAS were used to identify 274 and 526 FT genes, respectively, in our study.

### 3.5. Prioritizing the Candidate Genes to Expose the Regulatory Network of FT

In the different environments and genetic backgrounds, QTLs/loci for the flowering time were varied, and it is tough to functionally validate the genes that were identified in almost all of the linkage groups [11,12,15,16,23,25,26,27,30,31,32,33,34,37,58] in *B. napus.* Therefore, several genes related to FT in rapeseed remain unexplored, though most previously reported studies have concentrated on the essential flowering genes such as *FT* [59,60], *FLC* [35,61,62], *FRI* [63], *CO* [36] etc. In our study, we identified a total of 14 highly expressed FT DEGs (Table 2 and Figure 5) of *B. napus* that were homologous to the 13 FT genes of *A. thaliana* (e.g., *FRI*, *FLC*, *COL1*, *phyA*, *COL1*, *SEP1*, *NF-YC10*) in four integrated and co-localized genomic regions after the integration of both GWAS approaches with the transcriptomic study.

Of these 14 FT genes, 3 are involved in vernalization, photoperiod, photoperiod, circadian clock, aging, and gibberellin pathways. The gene *BnaA03g13320D* was homologous to *FRIGIDA (FRI)*, activated the *FLC*, and was discovered in the ~395 kb distal end of the SNP Bn-A03-p7177776 (6.45 Mb) (FT.A03.1) and also regulated trait budding and DTF. By intermingling with the histone methyltransferase *EFS* in vernalization, *FRI* promoted the up-regulation of *FLC* [64]. *B. napus* contains multiple copies of *FLC*, and several homologs are associated with FT variation [65]. In particular, *FLC* locus and its chromatin-mediated regulators are the most promising targets for regulating the flowering time of *B. napus* and has been stated to be a main transcriptional regulator that delays flowering by inhibiting the expression of floral integrators such as *FT*, *FD*, and *SOC1*, as previously reported in *A. thaliana* [66]. In this study, the gene *FLC* homolog to *BnaA03g13630D*, located in the ~210 kb proximal end of the integrated region FT.A03.1, is a key gene in the vernalization pathway, functioning as an inhibitor of flowering by binding to the *SOC1* promoter and regulating flowering time in *A. thaliana* [67]. Interestingly, in our GWAS result, *BnaA10g22080D* (*BnFLC.A10*) was detected in the ~333–391 kb upstream of the three SNPs, i.e., Bn-A10-p14657534, Bn-scaff_17109_2-p516588, and Bn-A10-p14711221, in chromosome A10. These findings were in the same location as those determined by Hou et al. (2012) [68]. Thus, *BnaFLC.A10*–*BnaA10g22080D* is an essential repressor of floral transition [69] and is likely a functional allele that is related to vernalization.

*Cycling DOF factor 2* (*CDF2*) encodes a Dof-type zinc finger domain-containing protein, which genetically controls flowering, and *cdf* mutant showed an early flowering phenotype [70]. Sun et al. (2015) also revealed *CDF2* in miRNA biogenesis regulated plant flowering at both the transcription and post-transcriptional levels [71]. We found its homologous candidate gene B*naA07g14740D* in the vicinity of environmentally specific QTL qFT.A07.1. *BnaC08g42660D* was detected as a *phyA (phytochrome A)* paralog located at the distal end of environmental specific hap.qFT.C08 (FT.C08) that controlled phenotype bolting and DTF. It also regulates circadian rhythms and flowering time [72] and plays essential roles in morning *FT* expression in the photoperiod pathway in *Arabidopsis* [73], which is probably a critical mechanism for photoperiodic flowering in nature.

## 4. Materials and Methods

### 4.1. Plant Materials and Field Experiments

A worldwide collection of 373 *B. napus* accessions was used for the GWAS analysis in the present study (Appendix A). The population phenotypic data was collected from three successive years (2013–2015) in Wuhan (East longitude 113°41′–115°05′, North 29°58′–31°22′ Latitude), which is located in the Hubei province in central China and from one year (2013) in Yangzhou (east longitude 119°01′to 119°54′, north latitude 32°15′to 33°25′), which is located in the Jiangsu province in the east of China. Every line was planted in a 2.5 m^2^ plot, and field tests followed a randomized design with three replications.

### 4.2. Evaluation and Statistical Analysis of Phenotypes

Different flowering stages comprising the time to budding (budding), time to bolting (bolting), and time of days to flowering (DTF, 50% of plants started flowering) stages were used to dissect the architecture of FT. The days from seeding to the development of buds and the achievement of a 3-cm-high main flower stalk were recorded as the budding and bolting stages, respectively. Lastly, the IBD trait was measured by the interval between bolting to DTF. The traits related to FT were evaluated in MEC and years.

The best linear unbiased estimates (BLUE) values were calculated across the different environments by considering genotypes as a fixed effect in the model through R package lme4 (CRAN-Package lme4 (r-project.org)) and lsmeans [74] and were then used to perform GWAS. By using QTL IciMapping V4.2 [75], the mean (days), standard deviation, skewness, kurtosis, two-way ANOVA, and broad-sense heritability (*h*^2^) values of the four flowering times traits in four environments of 373 *B. napus* accessions were measured. The frequency distributions of the four FT-related traits were constructed with the “Minitab” software. The pairwise correlation of coefficients for all of the traits was calculated using the package “PerformanceAnalytics” (https://CRAN.R-project.org/package=PerformanceAnalytics accessed on 12 April 2021) in R (https://www.R-project.org/ accessed on 12 April 2021).

### 4.3. SNP Genotyping and Quality Control

The updated CTAB method was used to extract genomic DNA from young leaves of 373 *Brassica napus* self-pollinated lines at the seedling level with three biological replicates [76]. The DNA was hybridized to a 60K Illumina Infinium HD Assay SNP array (Illumina Inc., San Diego, CA, USA) to obtain genotypes for 52,157 SNPs in each line. To guarantee the quality of SNP genotyping, in addition to biological and technological repeats, the SNPs were filtered based on the call frequency ≥ 85%, minor allele frequency ≥ 5%, cluster separation scores ≥ 0.4, heterozygosity ≤ 15%, and unique physical position in the *Brassica napus* “*Darmor-bzh*” reference genome (version 4.1) [2].

### 4.4. Genome-Wide Association Analysis

#### 4.4.1. SNP-Based GWAS

“EMMAX” (beta version) [77] was used to analyze the SNP-GWAS data. The matrix of the pairwise genetic distances, which was calculated using EMMAX, was used as the variance–covariance matrix of random effects to test the trait–SNP associations in an MLM model that was previously reported to be an optimal model for GWAS [42,44]. Significant *p*-value thresholds (*p* < 10^−5^) were set to control the genome-wide type 1 error rate, which was calculated by −log_10_ (1/n) (where, n = total SNPs) after rounding ≈ 4.35. The GWAS results were represented in a Manhattan plot using the recently developed package “*CMplot*” (https://github.com/YinLiLin/R-CMplot April 15, 2021) in R.

#### 4.4.2. Haplotype Based GWAS

The Haplotype-based GWAS (hap-GWAS) data were analyzed using the plink software package [78]. The haplotype blocks (HBs) were constituted using the following parameters: “--blocks no-pheno-req --blocks-max-kb 1000 --blocks-min-maf 0.05 --blocks-strong-lowci 0.70 --blocks-strong-highci 0.98 --blocks-recomb-highci 0.90 --blocks-inform-frac 0.95”. Haplotype association analysis was performed with the following parameters: “plink –file –hap --hap-assoc --allow-no-sex –noweb --out”. The significant thresholds (*p* < 3.65 × 10^−4^) for the Hap-GWAS data were set the same as those for SNP-GWAS data.

When a QTL can be identified in at least two different environments, then it is termed as being an environmentally stable QTL, and it can be an environmentally specific QTL when the QTL can be identified in at least two different consecutive years in the same environment. It can be considered a QTL hotspot/cluster if it can control at least two traits that are identified in more than one environment and is found at least three QTLs in the same chromosome. Here, qFT means “q”--QTL and “FT”-flowering time, and hap.qFT means hap-haplotype.

### 4.5. Comparison of QTLs Targeting Flowering Time in the Present Study with Previously Reported Study

To identify the novel QTLs and to confirm the reproducibility of QTLs in the present study, the loci associated with the FT in *B. napus* [11,12,14,15,16,23,25,26,27,30,31,32,33,34,37,58] that we had previously reported upon were collected to assess the co-localized intervals. We used the same reference genome sequence of *B. napus* (*Darmor-bzh*) [2] to align their physical positions (including the target intervals).

### 4.6. RNA-Seq/Transcriptomics Analysis

The SAM tissues of two early flowering accessions (AH110 and AH275) and two late-flowering accessions (AH218 and AH245) with two biological replicates were collected at the bolting stage. An RNA Prep Pure Plant Kit (TIANGEN, Beijing) was used to isolate total RNA, which was then used to build the RNA-seq library. The BGISEQ platform was used to sequence the well-constructed libraries. Adapters and reads with over 5% of non-sequenced (N) bases or more than 20% Q ≤ 20 were removed by SOAPnuke [79]. Clean readings were aligned to the reference genome using HISAT2 [80], and the rate of gene alignment was measured. The fragments per kilobase transcript per million reads (FPKM) values were calculated using Stringtie to calculate the expression levels of the genes and the transcripts [81]. The RNA-Seq reads were calculated through the R package “subread” [82]. DEGs were discovered using RNA-Seq reads through the DEseq2 method [83], which had an FPKM (fold-change) ≥1.00 and an adjusted *p*-value ≤ 0.05 [83].

### 4.7. Identification of Candidate Genes and Analysis of Their Expression Patterns

All of the annotated gene sequences from the candidate regions were downloaded from the *B. napus* reference genome, “Darmor-*bzh”* (http://www.genoscope.cns.fr/brassicanapus/ accessed on 25 April 2021) and were aligned against the genome sequence of *Arabidopsis* by local BLAST analysis. First, *B. napus* with the highest sequence similarity to genes in *Arabidopsis* were assigned as orthologous genes. Then, the *Arabidopsis* genes annotated as associated with FT traits were retrieved. Finally, their *B. napus* homologs within the one (1) Mb region of the target QTL were assigned as putative candidate genes for the QTL. A heat map was constructed through “TBtools” [84] using the normalized expression values (FPKM) of candidate genes.

## 5. Conclusions

This study explored the phenotypes of four traits related to the flowering time of *B. napus* in different environments. We performed a GWAS of both the SNP and haplotypes of four traits based on 60K SNP array genotypes of 373 *B. napus* accessions, and 61 SNP and 80 haplotypes significantly associated with these traits were detected on all chromosomes using the MLM model. Highly favorable alleles for promoting flowering time, vegetative (budding and bolting), reproductive (DTF), and the interval between bolting and DTF (IBD) were excavated. Moreover, we identified 14 FT candidate genes by combining GWAS and the transcriptomics study controlling these traits, which can be used to achieve an early maturing variety for future genetic engineering purposes to achieve an early maturing variety. In conclusion, we presented a series of investigative analyses of loci and candidate genes related to FT based on GWAS, which showed great power in uncovering genetic variation in flowering time in *B. napus* and improving our knowledge of the molecular mechanisms controlling flowering in rapeseed. The exclusive alleles that contribute to FT in *B. napus* can be directly applied to targeted marker-assisted breeding in rapeseed.

## Figures and Tables

**Figure 1 plants-10-02475-f001:**
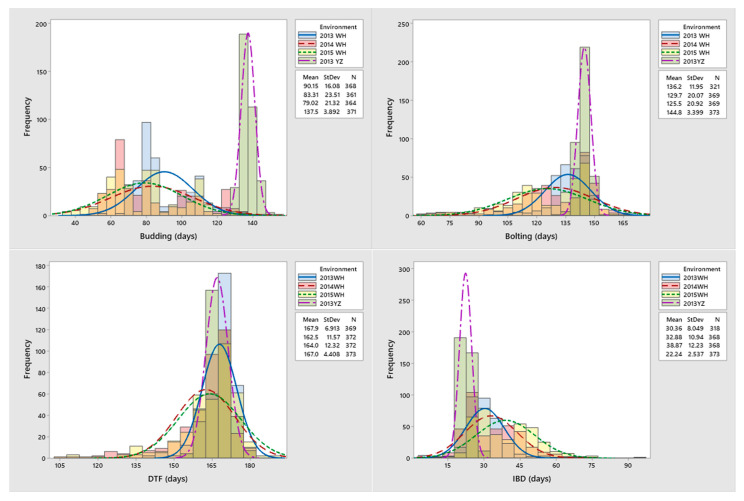
Frequency distributions of four flowering time traits in four environments. Where, the *x*-axis indicates the flowering time traits (days), and the *y*-axis indicates the frequency distributions.

**Figure 2 plants-10-02475-f002:**
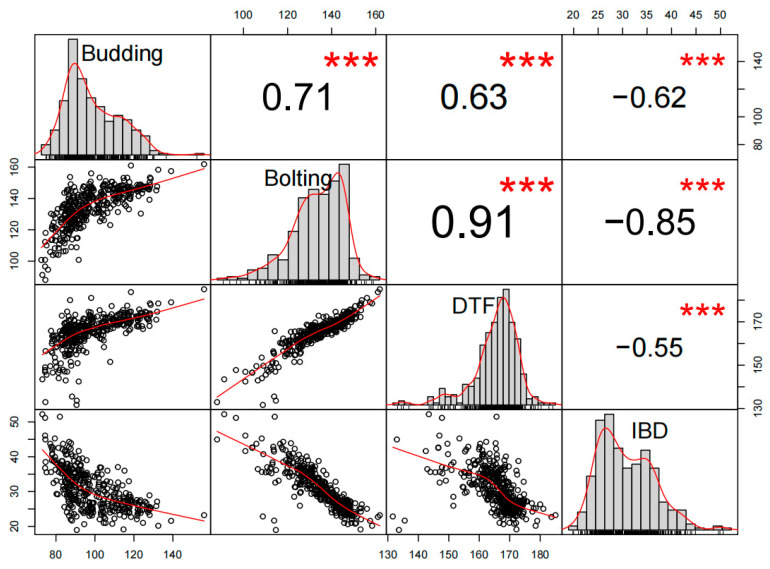
Correlation between budding, bolting, DTF, and IBD. *** Represents a correlation coefficient with the significance level (*p* < 0.001).

**Figure 3 plants-10-02475-f003:**
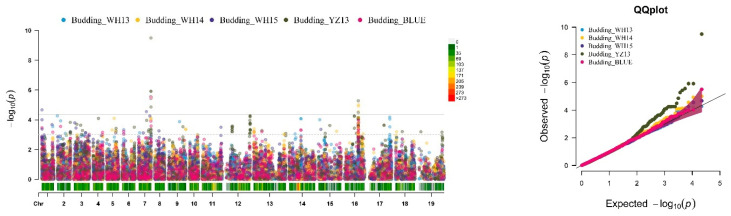
Genome-wide association scan based on SNP for four flowering-related traits (budding, bolting, DTF, and IBD). Left panel: Manhattan plots. The dashed horizontal line signifies the threshold for significant associations. The *x*-axis shows the physical position of all SNPs across *B. napus* chromosomes, and the *y*-axis shows the negative log_10_-transformed *p*-values for each association. Right panel: quantile–quantile (Q–Q) plots for four traits.

**Figure 4 plants-10-02475-f004:**
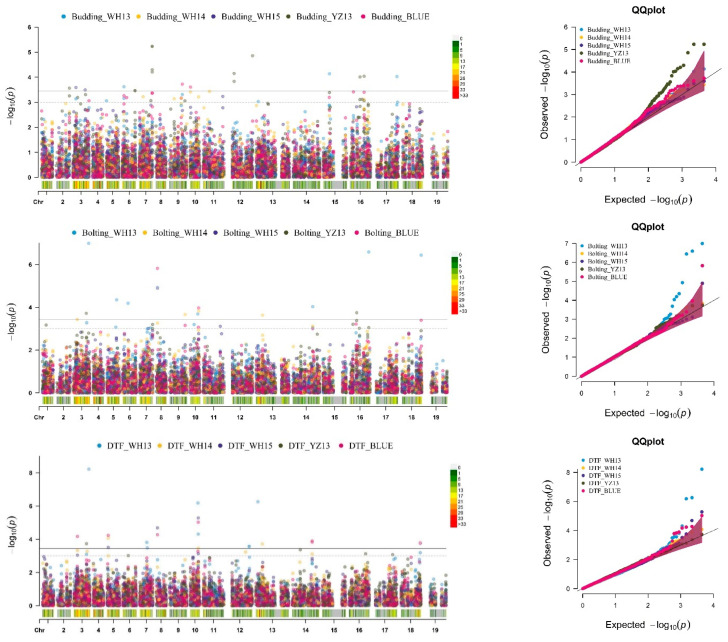
Genome-wide association scan based on haplotypes for four flowering-related traits (budding, bolting, DTF, and IBD). Left panel: Manhattan plots. The dashed horizontal line signifies the threshold for significant associations. The *x*-axis shows the physical position of all haplotypes across *B. napus* chromosomes, and the *y*-axis shows the negative log_10_-transformed *p*-values for each association. Right panel: quantile–quantile (Q–Q) plots for four traits.

**Figure 5 plants-10-02475-f005:**
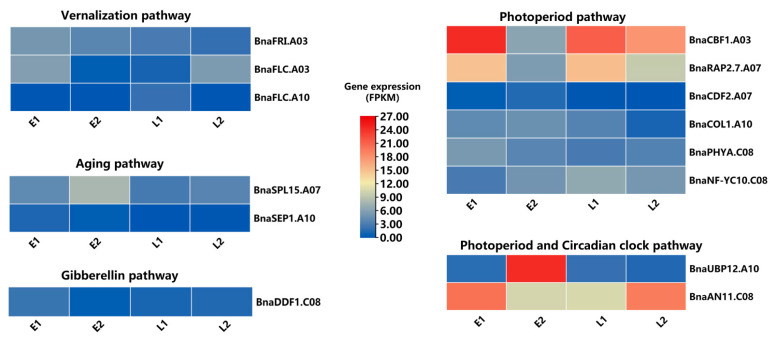
Highly expressed candidate genes in different flowering time pathways. The heatmap representing the expression levels (FPKM) in four samples (early and late). Here, E = early flowering samples and L = late-flowering samples.

**Table 1 plants-10-02475-t001:** Phenotypic variation of four flowering time traits in four environments.

Environment	Trait	Mean (Days)	SD	CV (%)	Kurtosis	Skewness	Heritability (*h*^2^)
2013WH	Budding	90.15	16.08	17.84	−0.03	0.81	0.90
	Bolting	136.22	11.95	8.77	6.37	−1.66	0.94
	DTF	167.89	6.91	4.12	11.09	−2.24	0.85
	IBD	25.88	13.09	50.58	0.98	−0.52	0.84
2014WH	Budding	83.31	23.51	28.22	−0.62	0.69	0.93
	Bolting	129.68	20.07	15.48	1.00	−1.02	0.92
	DTF	162.49	11.57	7.12	4.91	−1.97	0.94
	IBD	32.44	11.51	35.48	0.37	0.36	0.88
2015WH	Budding	79.02	21.32	26.98	−0.61	0.39	0.98
	Bolting	125.47	20.92	16.68	0.02	−0.75	0.98
	DTF	164.00	12.32	7.51	4.67	−1.79	0.97
	IBD	38.35	12.94	33.75	0.80	0.37	0.96
2013YZ	Budding	137.50	3.89	2.83	1.57	0.84	0.57
	Bolting	144.79	3.40	2.35	0.90	−0.43	0.53
	DTF	167.03	4.41	2.64	0.12	0.14	0.64
	IBD	22.24	2.54	11.41	4.39	−0.56	0.73

Where, 2013WH—Wuhan in 2013; 2014WH—Wuhan in 2014; 2015WH—Wuhan in 2015; 2013YZ—Yangzhou in 2013; SD—standard deviation; CV—coefficient of variation; Heritability—broad-sense heritability.

**Table 2 plants-10-02475-t002:** List of candidate genes related to flowering time.

Integrated/ Co-localized region	B napus gene ID	Gene Name	AT Ortholog	Gene Annotation	SNP QTL	Hap-QTL	Traits	Environment	SE1vSL1	SE1vSL2	SE2vSL1	SE2vSL2
FT.A03.1	BnaA03g13320D	FRI	AT4G00650.1	FRIGIDA	qFT.A03.1	hap.qFT.A03.1	Budding, DTF	WH15, BLUE	0.6464	1.3505	0.2236	0.9277
	BnaA03g13620D	CBF1	AT4G25490.1	C-repeat/DRE binding factor 1					0.0628	0.5349	−1.8762	−1.4040
	BnaA03g13630D	FLC	AT5G10140.1	FLOWERING LOCUS C					2.3706	0.0129	−0.6333	−2.9910
FT.A07.1	BnaA07g13990D	RAP2.7	AT2G28550.3	Related to AP2.7	qFT.A07.1	hap.qFT.A07.1	Budding, Bolting, IBD	WH13, WH15	−0.0614	0.4684	−1.4487	−0.9188
	BnaA07g14740D	CDF2	AT5G39660.2	Cycling DOF factor 2					4.8967	5.7036	5.9862	6.7932
	BnaA07g17550D	SPL15	AT3G57920.1	Squamosa promoter binding protein-like 15					0.3184	−0.1571	1.3920	0.9165
FT.A10	BnaA10g18420D	COL1	AT5G15850.1	CONSTANS-like 1	qFT.A10	hap.qFT.A10.3	Bolting, DTF, IBD	WH13, WH14, WH15, YZ13, BLUE	−0.0295	1.7775	0.5624	2.3694
	BnaA10g18480D	(SEP1)	AT5G15800.1	SEPALLATA1					5.7243	5.5249	4.5259	4.3265
	BnaA10g22080D	FLC	AT5G10140.1	FLOWERING LOCUS C					−7.3033	−3.2291	−6.9579	−2.8836
	BnaA10g24300D	UBP12	AT5G06600.1	Ubiquitin-specific protease 12					−0.5266	0.3302	3.2339	4.0907
FT.C08	BnaC08g40840D	ATAN11	AT1G12910.1	ANTHOCYANIN11		hap.qFT.C08	Bolting, DTF	WH13, WH15, BLUE	1.0874	−0.0768	−0.0832	−1.2475
	BnaC08g41070D	DDF1	AT1G12610.1	DWARF AND DELAYED FLOWERING 1					0.9130	0.6218	−1.3937	−1.6849
	BnaC08g42660D	PHYA	AT1G09570.1	Phytochrome A					1.0181	0.7956	0.4025	0.1801
	BnaC08g43430D	NF-YC10	AT1G07980.1	Nuclear factor Y, subunit C10					−1.2312	−0.7080	−0.3565	0.1666

## Data Availability

The data presented in this study are available in the article and Appendix A.

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
