# Peer review of "SNP- and Haplotype-Based GWAS of Flowering-Related Traits in Brassica napus"

_plants, 2021, doi:10.3390/plants10112475_

Round 1
Reviewer 1 Report
The manuscript is about QTL mapping for flowering date and traits related with days to flowering using GWAS approach. The manuscript may have scientific merit if published but the following issues must be addressed before accepting it for publication.
- Abbreviation should not be used in the "Abstract".
- What is opportune flowering time mean? define it.
- RCBD design is not for evaluating such large number of entries in the field. Row-Column design or other version of lattice design should have been used. Justify why RCBD?
- Revise line 565, 571-576, 146-147, 165-166, 189-190, 191-215, 231-239, and 249-255, 376-378. Please, rewrite the Results portion. It is poorly written. Move literature review to the intro part. Present only relevant information. Hard to follow in the current form.
Author Response
Reviewer-1:
The manuscript is about QTL mapping for flowering date and traits related with days to flowering using the GWAS approach. The manuscript may have scientific merit if published, but the following issues must be addressed before accepting it for publication.
Response: We appreciate your comments and suggestions that help us to improve the MS.
Comments-
- Abbreviation should not be used in the "Abstract".
Response: We corrected all of these according to reviewer comments.
- What is opportune flowering time mean? Define it.
Response: Opportune flowering time means appropriate timing of flower, which is crucial for survival in specific environments and regulates the life-cycle duration, yield, seed quality, resistance to diseases, and crop rotations systems.
- RCBD design is not for evaluating such a large number of entries in the field. Row-Column design or other versions of lattice design should have been used. Justify why RCBD?
Response: I am sorry for the misunderstanding about the RCBD. I corrected it with a randomized design with three replicates (Lu et al., 2019) instead of RCBD.
Reference:
Lu, K., Wei, L., Li, X., Wang, Y., Wu, J., Liu, M., Zhang, C., Chen, Z., Xiao, Z., Jian, H., Cheng, F., Zhang, K., Du, H., Cheng, X., Qu, C., Qian, W., Liu, L., Wang, R., Zou, Q., Ying, J., Xu, X., Mei, J., Liang, Y., Chai, Y.R., Tang, Z., Wan, H., Ni, Y., He, Y., Lin, N., Fan, Y., Sun, W., Li, N.N., Zhou, G., Zheng, H., Wang, X., Paterson, A.H., and Li, J. (2019). Whole-genome resequencing reveals Brassica napus origin and genetic loci involved in its improvement. Nat Commun 10, 1154.
- Revise line 565, 571-576, 146-147, 165-166, 189-190, 191-215, 231-239, and 249-255, 376-378. Please, rewrite the Results portion. It is poorly written. Move literature review to the intro part. Present only relevant information. Hard to follow in the current form.
Response: All corrected, according to the reviewer suggestion.
Reviewer 2 Report
Dear Authors,
Here are attached review comments-

Author Response
Reviewer-2:
The manuscript plants-1416699 explains and investigate flowering pathways at genome to phenome and transcriptomic level. However, there are a few minor typos and errors which must be revised with a little bit more details. After minor revision, I would like to recommend for the publication.
Response: We appreciate your comments and suggestions that help us to improve the MS.
Comments-
- Abstract is concise and well written except a few grammatical errors.
Response: Thanks for your positive comments. We corrected all of these according to reviewer comments.
- In Introduction, line 66-67, it requires a reference to quote the first study.
Response: We added the reference according to the reviewer suggestion.
- Materials and Methods are well written and explained well with description of tools and technologies.
- Results and discussion are very well explained with nicely describe flowering pathways in B. napus supporting with RNA-seq analysis.
Response: Thanks for your positive comments.
- Conclusion needs more explanatory details for geneticist and molecular breeders in order to develop early maturing B. napus varieties.
Response: We improved according to the reviewer suggestion.